# New Antibacterial Peptides from the Freshwater Mollusk *Pomacea poeyana* (Pilsbry, 1927)

**DOI:** 10.3390/biom10111473

**Published:** 2020-10-23

**Authors:** Melaine González García, Armando Rodríguez, Annia Alba, Antonio A. Vázquez, Fidel E. Morales Vicente, Julio Pérez-Erviti, Barbara Spellerberg, Steffen Stenger, Mark Grieshober, Carina Conzelmann, Jan Münch, Heinz Raber, Dennis Kubiczek, Frank Rosenau, Sebastian Wiese, Ludger Ständker, Anselmo Otero-González

**Affiliations:** 1Center for Protein Studies, Faculty of Biology, University of Havana, 25 street, 10400 Havana, Cuba; mgonzalez@fbio.uh.cu (M.G.G.); julio.perez@fbio.uh.cu (J.P.-E.); 2Core Facility for Functional Peptidomics, Faculty of Medicine, Ulm University, 89081 Ulm, Germany; armando.rodriguez-alfonso@uni-ulm.de; 3Core Unit of Mass Spectrometry and Proteomics, Faculty of Medicine, Ulm University, 89081 Ulm, Germany; sebastian.wiese@uni-ulm.de; 4Reference Center for Research and Diagnosis, Pedro Kourí Institute for Tropical Medicine, 11400 Havana, Cuba; annia@ipk.sld.cu (A.A.); antonivp@ipk.sld.cu (A.A.V.); 5General Chemistry Department, Faculty of Chemistry, University of Havana, Zapata y G, 10400 Havana, Cuba; femvicente@gmail.com; 6Synthetic Peptides Group, Center for Genetic Engineering and Biotechnology, P.O. Box 6162, 10600 Havana, Cuba; 7Institute of Medical Microbiology and Hygiene, University Hospital Ulm, 89081 Ulm, Germany; Barbara.Spellerberg@uniklinik-ulm.de (B.S.); Steffen.stenger@uniklinik-ulm.de (S.S.); mark.grieshober@uniklinik-ulm.de (M.G.); 8Institute of Molecular Virology, Ulm University, Meyerhofstrasse 1, 89081 Ulm, Germany; carina.conzelmann@uni-ulm.de (C.C.); jan.muench@uni-ulm.de (J.M.); 9Institute of Pharmaceutical Biotechnology, Ulm University, 89081 Ulm, Germany; Heinz.Raber@uni-ulm.de (H.R.); dennis.kubiczek@uni-ulm.de (D.K.); frank.rosenau@uni-ulm.de (F.R.)

**Keywords:** antimicrobial peptides, *Pomacea poeyana*, antibacterial activity

## Abstract

Antimicrobial peptides (AMPs) are biomolecules with antimicrobial activity against a broad group of pathogens. In the past few decades, AMPs have represented an important alternative for the treatment of infectious diseases. Their isolation from natural sources has been widely investigated. In this sense, mollusks are promising organisms for the identification of AMPs given that their immune system mainly relies on innate response. In this report, we characterized the peptide fraction of the Cuban freshwater snail *Pomacea poeyana* (Pilsbry, 1927) and identified 37 different peptides by nanoLC-ESI-MS-MS technology. From these peptide sequences, using bioinformatic prediction tools, we discovered two potential antimicrobial peptides named Pom-1 (KCAGSIAWAIGSGLFGGAKLIKIKKYIAELGGLQ) and Pom-2 (KEIERAGQRIRDAIISAAPAVETLAQAQKIIKGG). Database search revealed that Pom-1 is a fragment of Closticin 574 previously isolated from the bacteria *Clostridium tyrobutyrium,* and Pom-2 is a fragment of cecropin D-like peptide first isolated from *Galleria mellonella* hemolymph. These sequences were chemically synthesized and evaluated against different human pathogens. Interestingly, structural predictions of both peptides in the presence of micelles showed models that comprise two alpha helices joined by a short loop. The CD spectra analysis of Pom-1 and Pom-2 in water showed for both structures a high random coil content, a certain content of α-helix and a low β-sheet content. Like other described AMPs displaying a disordered structure in water, the peptides may adopt a helical conformation in presence of bacterial membranes. In antimicrobial assays, Pom-1 demonstrated high activity against the Gram-negative bacteria *Pseudomonas aeruginosa* and moderate activity against *Klebsiella pneumoniae* and *Listeria monocytogenes*. Neither of the two peptides showed antifungal action. Pom-1 moderately inhibits Zika Virus infection but slightly enhances HIV-1 infectivion in vitro. The evaluation of cell toxicity on primary human macrophages did not show toxicity on THP-1 cells, although slight overall toxicity was observed in high concentrations of Pom-1. We assume that both peptides may play a key role in innate defense of *P. poeyana* and represent promising antimicrobial candidates for humans.

## 1. Introduction

Antimicrobial peptides (AMPs) or host defense peptides (HDPs) are compounds with molecular masses within the 1–10 kDa range, of cationic (+2 to +9), hydrophobic and amphipathic character at physiologic pH [1]. They are part of the innate immune system, and their expression can be induced by bacterial components; therefore, environmental-related differential exposures to different pathogens can influence the AMP repertoire of hosts [2]. They display a broad spectrum of activities as they are active against Gram-positive and Gram-negative bacteria as well as fungi, viruses and parasites [3]. In general, AMPs show the ability to directly kill microbes, but they also modulate the immune system by orchestrating a local or systemic anti-infective action mainly in organisms with acquired immunity [4].

AMPs have been isolated from different sources due to their presence in all living organisms. Mollusks may represent an important source of AMPs because they rely solely on innate defenses to fight infections [5]. In addition, they are one of the largest Phyla with approximately 120,000 species, and many live in close proximity to human environments where anthropic transformation and contamination can influence the type and amount of microbial challenge conditioning their immune responses. One of the first findings of AMPs in mollusks is described in the work of Yamazaki et al. (1990) about *Aplysia kurodai* [6].

Due to the mentioned properties, AMPs have received significant attention as an alternative for treatment of infectious diseases, especially in recent years when the emergence of microbial strains resistant to conventional antibiotics became a serious health problem worldwide. According to the World Health Organization (WHO), antimicrobial resistance is one of the three biggest threats to human health [7]. In this context, the search for new effective antimicrobial agents is mandatory.

This study aims to identify new antimicrobial peptides from the snail *Pomacea poeyana.* We found Pom-1 and Pom-2, which were synthesized and functionally characterized. Both peptides showed high activity against *Pseudomonas aeruginosa* with low cell toxicity. These peptides could be used to generate new derivatives with a great therapeutic potential. In addition, this is the first report of the presence of AMPs in *P. poeyana*.

## 2. Materials and Methods

### 2.1. Invertebrate Collection and Sample Preparation

We sampled 15 adult specimens (40 mm shell length) of the freshwater ampullariid snail *Pomacea poeyana* (Pilsbry, 1927) in Havana, Cuba. All specimens were frozen before separating the shell from the body of the snails. Soft parts (foot and visceral mass) were homogenized in phosphate-buffered saline (PBS) using a blender followed by centrifugation at 10,000 rpm for 15 min at 4 °C. Supernatant was ultrafiltered with a cut-off of 10 kDa at 5371 rpm for 10 min at 4 °C. Subsequently, the low-molecular-weight fraction was lyophilized. The species used in this study is an endemic Cuban Ampullariidae, which is not endangered nor subjected to any regulation and occurs in large populations throughout Cuba [8].

### 2.2. Microorganism Strains and Growth Conditions

Three bacterial species were used for the biological assays: *Pseudomonas aeruginosa* (ATCC 29213), *Listeria monocytogenes* (ATCC BAA-679/EGD-e) and *Klebsiella pneumoniae* (ATCC 70063), obtained from the Institute of Medical Microbiology and Hygiene, University Clinic of Ulm, University of Ulm, Germany. Bacteria were cultured at 37 °C/5% CO_2_ overnight as preculture inoculum in Todd–Hewitt yeast medium.

Two Candida species were used: *Candida albicans* (ATCC 90028) and *Candida parapsilosis* (ATCC 22019), which were obtained from the Laboratory of Medical Mycology, IPK. Sabouraud dextrose medium was used for preculture inoculum at 37 °C for 48 h.

### 2.3. Cell Culture

hMDMs (human monocyte derived macrophages): Peripheral blood mononuclear cells (PBMCs) were isolated from human buffy coat via high density gradient centrifugation (Ficoll-Paque Plus; GE Healthcare, Munich) and monocytes were then purified from the PBMCs through adherence. The cells were then stimulated with granulocyte macrophage colony-stimulating factor (GM-CSF), 10 ng/mL (Miltenyi Biotec, Bergisch Gladbach) for 6 days at 37 °C, 5% CO_2_ in RPMI (Roswell Park Memorial Institute) 1640 medium (GIBCO, Invitrogen, Munich) supplemented with l-glutamine, 2 mmol/L (PAN Biotech, Aidenbach), HEPES (4-(2-hydroxyethyl)-1-piperazineethanesulfonic acid) 10 mmol/L (Biochrom, GmbH, Berlin), penicillin/streptomycin (100 U/mL/100 µg/mL) (Biochrom, GmbH, Berlin).

Vero E6: *Cercopithecus aethiops*-derived epithelial kidney (ATCC^®^ CRL-1586™) cells were grown in Dulbecco’s modified Eagle’s medium (DMEM) supplemented with 2.5% heat-inactivated fetal calf serum (FCS), 2 mM L-glutamine, 100 units/mL penicillin, 100 µg/mL streptomycin, 1 mM sodium pyruvate, and nonessential amino acids. Cells were grown at 37 °C in a 5% CO_2_ humidified incubator. 

TZMbl: TZMbl cells (HeLa cell derivative) were cultured in Dulbecco’s modified Eagle’s medium (DMEM) supplemented with 10% heat-inactivated fetal calf serum (FCS), 2 mM l-glutamine, 100 units/mL penicillin and 100 µg/mL streptomycin. Cells were grown at 37 °C in a 5% CO_2_ humidified incubator.

### 2.4. Sample Fractionation and Peptide Sequencing Analysis by Nanolc-ESI-MS-MS

Lyophilized samples were dissolved in 0.1% TFA/water, reduced with 5 mM DTT + 50 mM NH_4_HCO_3_ for 20 min at RT and subsequently alkylated with iodoacetamide for 20 min at 37 °C. The sample (15 µL) was analyzed using an LTQ Orbitrap Velos Pro system (Thermo Fisher Scientific) online coupled to an U3000 RSLCnano (Thermo Fisher Scientific) UPLC as described previously [9], with the following modifications: for sample fractionation, a binary gradient consisting of solvent A (0.1% FA) and solvent B (86% ACN, 0.1% FA) was employed. After loading onto the precolumn, the sample was concentrated and washed in 5% B for 5 min. In the first elution step, the percentage of B was raised from 5 to 15% in 5 min, followed by an increase from 15 to 40% B in 30 min. The column was washed with 95% B for 4 min and re-equilibrated for subsequent analysis with 5% B for 19 min.

Database search was performed using MaxQuant version 1.6.3.4 (https://www.maxquant.org/) [10]. For peptide identification, the built-in Andromeda search engine [11] correlated MS/MS spectra with the UniProt-reviewed Mollusca proteins (www.uniprot.org) and the APD3 antimicrobial peptide database http://aps.unmc.edu/AP/ [12]. Carbamidomethylated cysteine was considered as a fixed modification and oxidation (M) as a variable modification. False discovery rates were set to 0.01 on both peptide and protein level.

### 2.5. Antimicrobial Activity Prediction

The peptide list exported from MaxQuant (Max Planck Institute of Biochemistry, Martinsried, Germany) analysis was converted into a FASTA file and introduced in the following antimicrobial peptide prediction servers: CAMPR3 (Biomedical Informatics Centre, Mumbai, India), http://www.camp3.bicnirrh.res.in/predict, [13]; AMP scanner vr.2 (Bioinformatics and Computational Biosciences Branch, National Institute of Allergy and Infectious Diseases, MD, USA), https://www.dveltri.com/ascan/v2/ascan.html [14]; and iAMPpred (ICAR-Indian Agricultural Statistics Research Institute, New Delhi, India), http://cabgrid.res.in:8080/amppred/server.php [15]. The results were merged and the prediction probability values from all servers were averaged for every sequence. A rank list was organized by decreasing order of prediction probability (Appendix A).

### 2.6. Peptide Synthesis

Pom-1 and Pom-2 were synthesized automatically in a 0.10 mmol scale using standard Fmoc solid phase peptide synthesis techniques with the microwave synthesizer Liberty blue (CEM GmbH, Kamp-Lintfort, Germany). A resin preloaded with the corresponding C-terminal amino acid was used and provided in the reactor and washed with dimethylformamide (DMF). The Fmoc protecting group was removed with 20% (*v*/*v*) piperidine in DMF and initialized with microwaves, followed by being washed with DMF. Amino acids were added in 0.2 mol equiv to the reactor, and then HBTU 2-(1H-benzotriazol-1-yl)-1,1,3,3-tetramethyluronium-hexafluorophosphate) in a 0 and 5 mol equiv was dosed into the amino acid solution, followed by the addition of 2 mol equiv of *N*,*N*-diisopropylethylamine (DIEA). The coupling reaction was done with microwaves in a few minutes, and then the resin was washed in DMF. These steps were repeated for all amino acids in the sequence. The last step of the last amino acid was the final deprotection. Once the synthesis was completed, the peptide was cleaved in 95% (*v*/*v*) trifluoracetic acid (TFA), 2.5% (*v*/*v*) triisopropylsilane (TIS) and 2.5% (*v*/*v*) H_2_O for 1 h. The peptide residue was precipitated and washed with cold diethyl ether (DEE) by centrifugation. The peptide precipitate was then allowed to dry under vacuum to remove residual ether, and the peptide was purified using reversed-phase preparative high-performance liquid chromatography (Waters GmbH, Eschborn, Germany) in an acetonitrile/water gradient under acidic conditions on a Phenomenex C18 Luna column (5 mm pore size, 100 Å particle size, 250 × 21.2 mm). Next, the peptide was lyophilized on a freeze-dryer (Labconco, Kansas City, MO, USA) for storage prior to use. The molecular mass of the pure peptide was verified by liquid chromatography–mass spectrometry (Waters GmbH, Eschborn, Germany).

### 2.7. Structural Prediction

The 3D structure of Pom-1 was predicted using ab initio modeling in the QUARK de novo protein structure prediction server (Department of Computational Medicine and Bioinformatics, University of Michigan, MI, USA) (https://zhanglab.ccmb.med.umich.edu/QUARK/) [16] with default parameters. The 3D structure of Pom-2 was predicted using a homology modeling method. Template search and model building was performed in the SwissModel server (SIB Swiss Institute of Bioinformatics, Basel, Switzerland) (https://swissmodel.expasy.org/) [17] with default parameters. The protein used as the template was the papiliocin isolated from *Papilio xuthus* (PDB: 2IA2) [18], with a sequence identity of 45.45% to Pom-2. Assessment of the predicted models was performed using the MolProbity server (http://molprobity.biochem.duke.edu/) [19]. Selected analysis included angle, length, clashscore, rotamer, Ramachandran and beta carbon evaluations. Helix hydrophobicity and hydrophobic moment were calculated using the totalizer module of the MPEx v3.3 program (Department of Physiology and Biophysics, Center for Biomembrane Systems, University of California, Irvine, CA, USA). [20].

### 2.8. Circular Dichroism Measurement

Pom-1 and Pom-2 solutions for circular dichroism (CD) measurement were prepared at 30 µM concentration in MilliQ^®^ (Merck Millipore, Burlington, MA, USA) water. CD spectra were recorded in a JASCO-1500 Circular Dichroism Spectrometer with a 260–180 nm measurement range, 5 nm/min scan speed, 0.1 mm path length, 0.2 data pitch and 2 s data integration time. Measured spectra were processed and deconvoluted using JASCO’s spectral analysis software and secondary structure estimation database.

### 2.9. Bioassays Against Bacteria

Antibacterial activity was evaluated using agar diffusion assay [21]. Bacteria were cultured at 37 °C/5% CO_2_ overnight, pelleted by centrifugation and washed in 10 mM sodium phosphate buffer. Following resuspension in 10 mM sodium phosphate buffer, optical density was determined at 600 nm and 2 × 10^7^ bacteria were seeded into a Petri dish in 1% agarose. After cooling at 4 °C for 30 min 3–5 mm holes were punched into the 1% agarose. Synthesized peptides, adjusted to the desired concentration in 10 µL of buffer, were filled into the agar holes. Following three-hour incubation at 37 °C, plates were overlaid with 1% tryptic soy agarose, solved in 10 mM sodium phosphate buffer. Inhibition zones (in cm) were measured following 16–18 h incubation time at 37 °C/5% CO_2_.

### 2.10. ^3^H-Uracil Proliferation Assay

Extracellular *Mycobacterium tuberculosis* (Mtb) (2 × 10^6^) were distributed into 96-well flat bottom plates (Nunc) in triplicates and incubated for three days at 37 °C/5% CO_2_. Next, 3H-Uracil (0.3 µCi/mL, Biotrend, Cologne, Germany) was added overnight at 37 °C/5% CO_2_. Mtb were then fixed and killed with 4% paraformaldehyde (PFA) at room temperature (RT) for 20 min and were then harvested (Cell harvester; Inotech, Reutlingen, Germany) onto a filtermat (Perkin Elmer, Waltham, MA, USA). Afterwards, wax plates (Meltilex A; Perkin Elmer) containing scintillation liquid were melted onto the mats. Samples were measured with a beta counter (Hidex sense micro beta counter), and the mean value per triplicate was calculated.

### 2.11. Antifungal Bioassays

Antifungal activity was determined according to the “Clinical and Laboratory Standards Institute” guidelines M27-A3 broth microdilution assay with modifications (turbidimetric detection) [22]. Based on cell density measurements, the minimal inhibitory concentration (MIC) was derived from a Lambert-Pearson plot [23]. Flat bottom sterile 96 well plates (SARSTEDT, AG and Co. KG, Nümbrecht, Germany), RPMI 1640 without sodium bicarbonate, MOPS-buffered (Sigma-Aldrich-Merck, Darmstadt, Germany) were used for the test, and readings were performed at λ = 600 nm.

### 2.12. Antiviral Bioassays

The ZIKV (Zika virus) infection rate was determined in Vero E6 cells. Therefore, 12,000 cells were seeded into 96-well plates. The next day, medium was removed and 70 µl fresh x-vivo medium (Lonza, Basel, Switzerland) and 10 µl of peptide titration series were added. Then, cells were infected with 20 µL of ZIKV MR766. Two days later, infection rates were determined with a cell-based ZIKV immunodetection assay described previously [24]. HIV infection was quantified by the reporter activity of TZMbl cells. Therefore, 10,000 cells were seeded. The next day, medium was removed and 70 µl fresh x-vivo medium (Lonza) and 10 µl of peptide titration series were added. Then, cells were infected with 20 µL of HIV-1 NL4-3. Two days later, the infection rate was determined by quantification of β-galactosidase expression according to Müller et al. [24]. All infection values were corrected for the background signal derived from uninfected cells, and untreated controls were set to 100% infection.

### 2.13. Cell Viability Assays

PrestoBlue™: In cell culture medium supplemented with 5% heat-inactivated human serum (PAN Biotech, Aidenbach, Germany), 1 × 10^5^ macrophages per well were distributed into a microplate (Nunclon™ Delta 96-Well MicroWell™ Plates, Sterile, Thermo Scientific, Dreieich, Germany). Controls were media control, heat-inactivated cells, and diluent control dimethylsulfoxide (DMSO; Carl Roth, Karlsruhe, Germany). Cells were then stimulated overnight with different concentrations of each of the synthesized peptides. Twenty microliters of PrestoBlue Cell Viability Reagent (Thermo Fisher Scientific Life Technologies GmbH, Darmstadt, Germany) per well was added and incubated for 20 min at 37 °C/5% CO_2_. The fluorescence was measured at the excitation wavelength of 560 nm and emission wavelength of 600 nm with the TECAN infinite M200 microplate reader (Tecan Group Ltd., Männedorf, Switzerland). The optical density (OD) of the media control was then subtracted from the other OD values. Unstimulated cells were set to 100% viability.

MTT (3-(4,5-dimethylthiazol-2-yl)-2,5-diphenyltetrazolium bromide)**:** Cells were treated in the same way as in the antiviral bioassays, but medium was used instead of virus inoculum. Two days later, metabolic activity was assessed by MTT-based assay. In brief, medium was removed, 100 µL MTT solution (1:10 diluted with PBS) was added and incubated at 37 °C for 3 h. Then, supernatant was discarded and formazan crystals dissolved in 100 µL 1:1 DMSO–ethanol solution. Absorption was measured at 490 nm and baseline corrected at 650 nm (Vmax Kinetic ELISA microplate reader). Values of untreated controls were set to 100% viability.

## 3. Results

### 3.1. Identification of Antimicrobial Peptides in Pomacea poeyana Extracts and Prediction of Antimicrobial Activity

We carried out a peptide extract of the Cuban freshwater snail *Pomacea poeyana* and analyzed it by chromatography in combination with mass spectrometry (LC-MS/MS). Thirty-seven peptide sequences were identified by MaxQuant processing of the MS raw data. From these 37 sequences, 33 were identified as protein fragments from molluscan species, whereas the other four peptide sequences as fragments of larger known antimicrobials (Appendix A). The peptide sequence hits in FASTA format were introduced into the AMP prediction servers CAMPR3, AMP scanner vr.2 and iAMPpred. The two highest scored peptide hits were selected for synthesis and further biological evaluation for their antimicrobial activity.

Database search revealed that Pom-1 is a fragment of Closticin 574, a class II 82-amino-acid bacteriocin produced by *Clostridium tyrobutyricum* ADRIAT 932 with antibacterial activity against *Clostridium beijerinckii* ATCC 25752, *Clostridium tyrobutyricum* NIZO B570, *Clostridium tyrobutyricum* NIZO B575, *Clostridium tyrobutyricum* CNRZ500, *Clostridium tyrobutyricum* NIZO B590, *Lactobacillus alimentarius* L4, *Lactobacillus buchneri* L4 and *Lactobacillus saké* ATCC 15521 [25]. Pom-2 is a fragment of 8.4.1, a 39-amino-acid cecropin D-like peptide isolated from *Galleria mellonella* hemolymph, which has activity against *Escherichia coli* [26].

### 3.2. Structural Prediction

A template search for Pom-1 sequence in the SwissModel server did not yield any homologous protein with more than 30% sequence identity to be used as a template. Without any suitable template, the structure of Pom-1 was predicted using ab initio prediction methods. The QUARK modeling server generated five models for Pom-1, named Pom-1_1 to Pom-1_5. The five models were very similar, comprising two alpha helices joined by a short loop. In all cases, the N-terminal helix was shorter than the C-terminal helix. The main difference among the models was the length of the N-terminal helix (ranging from 12 aa in Pom-1_1 to 9 aa in Pom-1_2), the angle between both helices and the residue rotamers.

The SwissModel server successfully created a model for Pom-2, using a papiliocin isolated from *Papilio xuthus* (PDB: 2IA2) as the template. The predicted structure consisted of two alpha helices joined by a short loop, similar to the structure of other known cecropins (Figure 1). The server model evaluation reported a global model quality estimate (GMQE) of 0.75 (1 is best) and a Q-mean Z-score of –1.47.

In order to assess the geometric quality of the generated models, all five models from both Pom-1 and Pom-2 were analyzed using the MolProbity server. The selected analysis included bond angle, length, clashscore, rotamer, Ramachandran and beta carbon evaluations. This structural analysis was also used to select one of the Pom-1 models as the best predicted structure. The evaluation results are displayed in Table 1 and the final selected models are highlighted in gray. Molprobity score: combination of the clashscore, rotamer and Ramachandran evaluations into a single score. Clashscore: number of < 0.4 Å steric overlaps per 1000 atoms. Poor rotamers: number of rotamers that are outside the bounds of a rotamer definition. Favored rotamers: number of rotamers that are close to the most favored rotamer conformations. Rama outliers/Rama favored: number of residues in the disallowed/favored regions in the Ramachandran analysis. Cβ deviations: Number of Cβ deviation of 0.25 Å or more from the ideal position. Bad bonds/angles: number of bonds length/angles that deviate more than 4 σ from average. A more detailed explanation of every evaluation can be found in the MolProbity webpage: http://molprobity.biochem.duke.edu/help/validation_options/validation_options.html. Pom-1_5 showed the best MolProbity score of all Pom-1 models, and thus was selected as the best structure among all generated models (Figure 1).

The hydrophobic moment of Pom-1 and Pom-2 N-terminal and C-terminal helices was calculated using the Totalizer module of the MPEx program (Figure 2). The N-terminal helix of Pom-1 displayed a low hydrophobic moment of 3.85 (0.35 per residue), whereas the C-terminal helix showed a high hydrophobic moment of 9.79 (0.70 per residue). In the case of Pom-2, both N-terminal and C-terminal helix displayed noticeable hydrophobic moments of 11.85 for the N-terminal helix (0.70 per residue) and 9.31 for the C-terminal helix (0.62 per residue). A helical wheel diagram of the four analyzed helices can be found in Figure 2.

As previously stated, all five predicted models for Pom-1 were very similar, with a tertiary structure consisting of two alpha helices joined by a hinge loop. The structure quality of the five models was assessed using the MolProbity server in order to (1) find whether any model has nonfavorable geometries due to poor modeling (2) select the model with most favorable evaluations as best predicted structure of Pom-1.

Most models displayed several steric clashes, Ramachandran outliers and bad angles, a common indicator of modeling problems. However, Pom-1_5 displayed none of these while having the best number of favored residues in the Ramachandran plot. The model also displayed a low number of poor rotamers and Cβ deviations, which altogether suggest a model without noticeable structural problems. The MolProbity score represents the central protein quality statistics, combining the clashscore, rotamer and Ramachandran evaluations into a single score. Again, Pom-1_5 displayed the best MolProbity score (1.99), far below the other models’ score, which ranged from 2.71 to 3.1. This means that Pom-1_5 structure quality is similar to an average 2.0 Å resolution protein in the protein data bank, which is an acceptable result. Overall, Pom-1_5 was considered and adequate model for representing Pom-1 structure.

In the case of Pom-2, a suitable template could be found and thus homology modeling was used. The SwissModel server evaluation of the generated model reported a GMQE of 0.75 (1 is best), and a Q-mean Z-score of –1.47 (above –4 is good, close to 0 is best), both indicators of an acceptable model. The Molprobity structure assessment revealed that almost all the residues lied in the Ramachandran favored area, with only one Ramachandran outlier (ALA18). No unusual bonds, angles or atomic clashes were observed, and the number of poor rotamers and Cβ deviations were low. Overall, the model shows good quality, represented by a low MolProbity score close to 1.4.

### 3.3. Circular Dichroism Spectroscopy

Pom-1 and Pom-2 displayed similar spectra with a deep minimum near 200 nm and a very small shoulder near 220 nm (Figure 3). The CD spectra analysis predicts high random coil content for both structures (48.1% for Pom-1 and 54.5% for Pom-2) and very low β-sheet content, with Pom-1 having the highest content of both α-helix and β-sheet (Table 2).

### 3.4. In Vitro Antimicrobial Activity

Antifungal activity of *P. poeyana* peptides was tested using microdilution assay against two species of *Candida* sp. Neither of these peptides showed activity against *Candida* sp. at 100 µg/mL. Antibacterial activity of *P. poeyana* peptides was tested using agar diffusion assay against Gram-positive and Gram-negative bacteria. Table 3 shows the antibacterial activity of all peptides against Gram-negative species *P. aeruginosa* and *K. pneumoniae* and Gram-positive species *L. monocytogenes*. Susceptibility to different antimicrobial peptides was tested by agar diffusion assay. Ten microliters of a stock solution at a concentration indicated above was spotted onto an agar containing the respective bacterial species. After overnight incubation at 37 °C, inhibition zones (cm) around the tested antimicrobial peptides (AMP) were measured. All tests were done in triplicate.

Pom-1 showed activity against all evaluated bacteria while Pom-2 exhibited antibacterial action only against *P. aeruginosa* and *L. monocytogenes* from 30 µg/mL. However, Pom-1 showed activity at the lowest concentration evaluated. The antimycobacterial activity of *P. poeyana* peptides on virulent extracellular Mtb was insignificant at 100 µg/mL.

Additionally, the activity of Pom-1 on ZIKV and HIV-1 infection was assessed. As shown in Figure 4, Pom-1 inhibits ZIKV infection slightly at concentrations up to 10 µg/mL but enhances HIV-1 infection slightly at concentrations of 20 µg/mL (Figure 4A). However, Pom-1 was cytotoxic on Vero E6 and TZMbl cells (Figure 4B) so the data should be interpreted with caution. Hence, the 50% cytotoxic concentration (CC_50_) and half maximal inhibitory concentration (IC_50_) were calculated by applying c(Inhibitor) vs. response—Variable slope (four parameters) in GraphPadPrism. The selectivity index (SI = CC_50_/IC_50_) thereof was calculated (Table 4), which allows evaluation of the specificity of the antiviral effect.

### 3.5. Cytotoxic Effect on Human Macrophages

The cytotoxic activity of *P. poeyana* peptides against primary human macrophages was evaluated. Figure 5 shows that Pom-1 reduced cell viability of primary human macrophages to 80% at 100 µg/mL, whereas Pom-2 is not cytotoxic.

## 4. Discussion

Antimicrobial resistance to conventional antibiotics has increased dramatically in the last decades and has become one of the most pressing global public health burdens worldwide. Therefore, it is necessary to find alternative antimicrobial agents. AMPs represent a potential option to develop new generations of antimicrobials agents [1]. They are very efficient antimicrobials found in all living forms [2].

Invertebrates are an important source of AMPs because they do not have an adaptative immune system and depend almost exclusively on their innate immune system, where AMPs play a crucial role [3]. Mollusks have been largely targeted as natural sources of AMPs partly due to their high diversity (only outnumbered by arthropods) and ecological plasticity, adapting to almost all types of habitats. Cuba is considered a hotspot of mollusk diversity worldwide, with nearly 95% of endemic species [8].

The present study reports the identification of 37 peptides from an endemic freshwater ampullariidae *P. poeyana,* known to occur in stable habitats (e.g., rivers, permanent ponds, lakes) and to resist a fair range of anthropogenic pollution [8,27]. Due to its relatively large size, it can be easily collected and manipulated. Our results constitute the first molecular characterization of this endemic snail, resulting in the first report of AMPs in this species.

From these 37 peptides, we selected two for synthesis and further biological evaluation. The selection was based on prediction probability values of antimicrobial activity, and those peptides exhibiting the highest score were selected: Pom-1 and Pom-2. These are fragments of larger proteins with demonstrated antibacterial activity. This result suggests that Pom-1 and Pom-2 can be induced through the cleavage of larger proteins from microorganisms found in the habitat of the mollusk. Similar findings have been reported in studies with other invertebrates, which show that AMP production is triggered by activation of different immune signaling pathways after recognition of molecules from microorganisms such as lipopolysaccharides (LPS) and L-glucans [28,29].

Pom-1 and Pom-2 do not belong to the group of prolin-rich AMPs from mollusks [30], nor do they contain disulfide bridges like the majority of AMPs from mollusks, many of them belonging to the family of defensins or are defensin-like peptides and larger proteins [31]. However, a few AMPs from mollusks exist that do not have cystein residues, e.g., the novel 55 amino acid peptide cgMolluscidin (extracted from the pacific oyster *Crassostrea giga*) [32], the piscidin-like AMPs (derived from *Oreochromis niloticus*) [33] and others.

When a protein does have an unknown 3D structure, homology modeling is usually a good choice for structural prediction. However, in order to produce reliable results, homology modeling requires an adequate template, i.e., a homologous protein with known structure and high sequence identity to the target. If no suitable template with more than 25% sequence identity can be found, homology modeling could produce inaccurate results [34]. In the case of Pom-1, the search for homologous proteins performed in the SwissModel server did not yield any suitable template protein with >25% sequence identity. In those cases, other structure prediction methods can still produce useful results. Ab initio approaches have seen some success in the prediction of the 3D structure of small proteins without using template proteins. [35]. The QUARK algorithm from Zhang Lab has ranked as one of the best ab initio prediction programs in several CASP experiments [36], and is readily available from its web server [1].

The predicted structure of Pom-1 and Pom-2 is highly helical in nature, with many positively charged residues and a similar tertiary structure arrangement of two alpha-helix joined by a short loop, a common feature in some antimicrobial peptides from insects such as sarcotoxin IA and Pd [37], melittin [38], papiliocin [39] and cecropins [40].

Pom-1 is a fragment of a class II bacteriocin, closticin 574, isolated from *Clostridium tyrobutyricum*. No 3D structure of this or any other related closticin is known [25]. The predicted structure is similar to cecropins, which share similar sequence length and lack of cysteine in their sequence. However, in Pom-1 the N-terminal and C-terminal helices are almost parallel, whereas in cecropins both helices have a bent angle of about 45–80° [18]. This could be because the QUARK modeling server uses a force field optimized for proteins in water, whereas most cecropin structures have been determined in phospholipidic micelles [18]. In water, the helix packing could favor hydrophobic contacts, but in micelles, the peptide could adopt an extended conformation for interacting with the membrane.

Pom-2 is a fragment of a cecropin D-like peptide, part of the well-known cecropin family. Its 3D structure displays two alpha helices joined by a flexible hinge region, with the amphiphilic N-terminal alpha helix containing a hydrophilic sector larger than the hydrophobic sector. Those are common features shared by many members of the cecropin family. The C-terminal helix, however, is also amphiphilic with a well-defined hydrophobic and hydrophilic sector. This contrast with other cecropins (including the papiliocin used as modeling template), which have a C-terminal helix with a larger number of hydrophobic residues than hydrophilic ones [18]. This feature could be related to differences in specificity against different pathogens.

The predicted secondary structure for both Pom-1 and Pom-2 peptides in water displayed a high content of unordered structures. This is in accordance with other described AMPs like cecropins and magainins, which display a disordered structure in water but adopt a helical conformation in presence of bacterial membrane analogues [41,42]. The CD spectra in water of Papiliocin, the peptide used as template for Pom-2 modeling, is very similar to Pom-1 and Pom-2 with high random coil content. However, in DPC micelles the papiliocin structure changed to a helix-hinge-helix conformation, as confirmed by NMR experiments [18]. This ordered conformation is very similar to the predicted structure of Pom-1 and Pom-2, and is theorized that the structure of both peptides could behave in a similar manner to papiliocin when changing environment from water to membranes.

In the antibacterial assays, the peptides showed a potent antibacterial action mainly against Gram-negative strains of *P. aeruginosa* and *K. pneumoniae*, but only Pom-1 had antibacterial action against Gram-positive *L. monocytogenes*. These bacterial strains are not multidrug-resistant microorganisms, but it is important to note that all of them are opportunistic pathogens that can cause severe infections. In addition, *P. aeruginosa* causes approximately 10% of infections acquired in hospitals [43] and *K. pneumoniae* causes a wide range of infections, including pneumonias, urinary tract infections, bacteremias and liver abscesses [44]. On the other hand, *L. monocytogenes* is a foodborne pathogen responsible for listeriosis, a disease associated with high mortality rates mainly because it can cause infection of the fetus in pregnant women. Also, it can cause meningitis, meningoencephalitis or febrile gastroenteritis in general population [45,46]. However, the emergence of resistant strains of these bacterial species has caused the search for alternatives for their treatment.

Another important bacterium is *M. tuberculosis,* the causal agent of tuberculosis. In recent years the treatment of this reemerging disease has been challenging due to multiresistant strains to this bacterium. In this situation, the search for new molecules to treat tuberculosis is a necessity [47]. However, *P. poeyana* peptides showed an insignificant action against *M. tuberculosis,* with percentages of activity below 20%.

On the other hand, Pom-1 and Pom-2 were not effective against *C. albicans* and *C. parapsilosis* at concentrations evaluated. Pom-1 slightly inhibits ZIKV infection and slightly enhances HIV-1 infectivity in vitro.

Pom-1 and Pom-2 share structural characteristics with some antibacterial peptides like sarcotoxins, papiliocin and melittin, mentioned above. They have demonstrated antibacterial activity against both Gram-positive and Gram-negative bacteria [40]. In this sense, Lee and collaborators demonstrated that papiliocin have bacterial cell selectivity with no cytotoxicity against NIH 3T3 cells and related this action with their high cationicity [40].

Other peptides with antibacterial activity are the cathelicidin family, which showed their strong effects on a large number of drug-resistant strains clinically isolated, even super-resistant bacteria. Its antibacterial activity is attributed to its N-terminal amphipathic α-helix [48].

Host cell toxicity is one of the factors that limit the clinical use of AMPs [49]. In this sense, we evaluated toxicity of *P. poeyana* peptides on human primary macrophages. These cells previously have been used for evaluating in vitro toxicity of AMPs. Both peptides did not present significant toxic effects on human primary macrophages, although Pom-1 showed a slight decrease in cell viability, approximately 30% at 100 µg/mL. However, this concentration is significantly higher than concentrations where the peptide showed antimicrobial action, so effective concentrations of Pom-1 seem to be nontoxic. According to Strandberg and collegues, smaller versions of larger peptides are less toxic at the same concentrations [50]. Keeping this in mind, the generation of shorter derivatives of Pom-1 could be an excellent strategy to reduce its toxicity.

Our findings suggest a selectivity of *P. poeyana* peptides for bacteria pathogens, mainly Gram-negatives, and it matches with antibacterial activity reported for Closticin 574 [26] and a cecropin D-like peptide [27]. The selectivity of AMPs is attributed to their affinities for the cytoplasmic membranes of different cells and is determined by physiochemical properties such as hydrophobicity (H), hydrophobic moment (uHrel), cationicity, amphipaticity etc. [51].

In this sense, both peptides have positive charges (+4 for Pom-1 and +2 for Pom-2) that favor their interactions with negative bacterial membranes and their antimicrobial activity. Bacteria have different anionic components in comparison with mammalian cells. The outer monolayer of bacterial membranes is composed essentially by lipids charge negatively like phosphatidylglycerol (PG) and cardiolipin (CL). In contrast, the outer monolayer of mammalian cells contains neutral lipids such as phosphatidylcholine (PC) and sphingomyelin (SM). In other sizes, cell wall (Gram-positive bacteria) or external membrane (Gram-negative bacteria) also contain anionic molecules such as lypoteichoic acids and lipopolysaccharides (LPS), respectively [52,53].

Previously, the existence of a threshold charge (usually +6) has been proposed, which facilitates anchoring of AMPs to the membrane, resulting in improved antibacterial properties [54]. However, Pom-1 has a higher net charge than Pom-2, suggesting that the first could be more active.

Another main parameter influencing peptide affinity for membranes is hydrophobicity (intrinsic capability of a peptide to move from an aqueous into a hydrophobic phase). Hydrophobicity not only affects the cell selectivity but also modulates the mode of peptide–membrane interaction and is positively correlated with cytotoxic activity [55]. Typically, AMPs present a hydrophobic percentage near 50%. Hydrophobicity of Pom-1 is 52.9%, whereas for Pom-2 it is 22%. According to these values, Pom-1 should be more active than Pom-2 but a little toxic to mammalian cells, as shown by the results of antimicrobial and cytotoxic activity assay (Section 3).

Amphipathicity is thought to be essential for selectivity and mechanism of action of AMPs, which in many cases involve membrane disruption of the target organism. This property is related to the distribution of polar and hydrophobic residues and can be quantified by the uHrel [56]. As shown by in silico prediction, both peptides have an amphiphilic alpha-helix structure with positively charged residues. According to the hydrophobic moment analysis, the N-terminal helix of Pom-1 is mostly hydrophobic and the C-terminal helix is amphiphilic. This combination of one hydrophobic and one hydrophilic helix is also common in many AMPs [18].

Structural similarity of both peptides with cecropins suggests that there could be similarities between their mechanisms of action, which for cecropins involves disruption of the bacterial membrane through a carpet model [40].

## 5. Conclusions

In summary, antimicrobial peptides (Pom-1 and Pom-2) identified from the mollusk *Pomacea poeyana* are amphiphilic alpha-helical peptides. They showed potent antibacterial activity against *P. aeruginosa* and a nonsignificant toxicity effect against mammalian cells. The current work demonstrated for the first time the presence of antimicrobial peptides in the freshwater mollusk *P. poeyana*. We assume that these peptides could be modified to generate new derivatives with improved therapeutic potential.

## Figures and Tables

**Figure 1 biomolecules-10-01473-f001:**
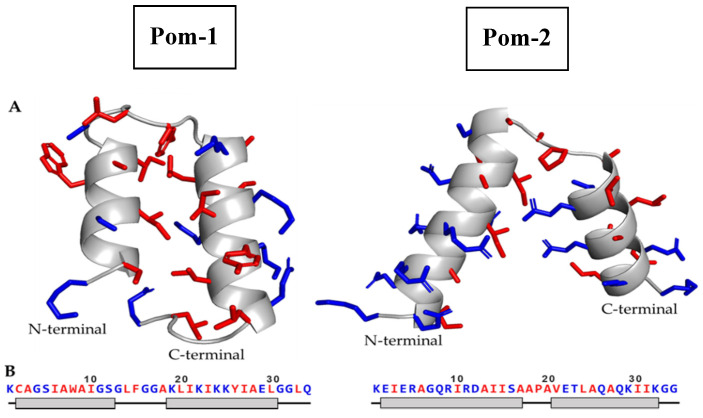
(**A**) Cartoon diagrams of the predicted structure for Pom-1 (left) and Pom-2 (right), modeled using QUARK and SwissModel servers (Manufacturer, City, State abbrev., if USA or Canada, Country), respectively. Residue side chains are represented in sticks. (**B**) Amino acid sequence of Pom-1 and Pom-2 models and its associated secondary structure. The gray boxes represent alpha helix residues; the black lines represent unordered residues. Hydrophobic residues are highlighted in red, hydrophilic residues in blue.

**Figure 2 biomolecules-10-01473-f002:**
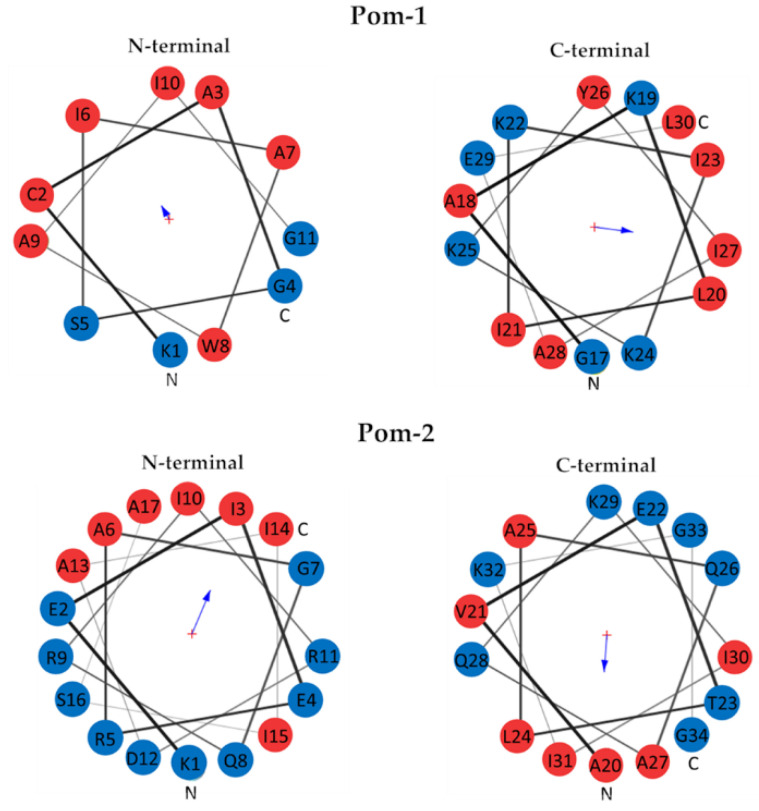
Helical wheel diagrams of N-terminal and C-terminal helix of Pom-1 (top) and Pom-2 (bottom). Hydrophobic residues are displayed in red, hydrophilic residues in blue. The arrow size is proportional to the hydrophobic moment and points to the hydrophobic side of the helix.

**Figure 3 biomolecules-10-01473-f003:**
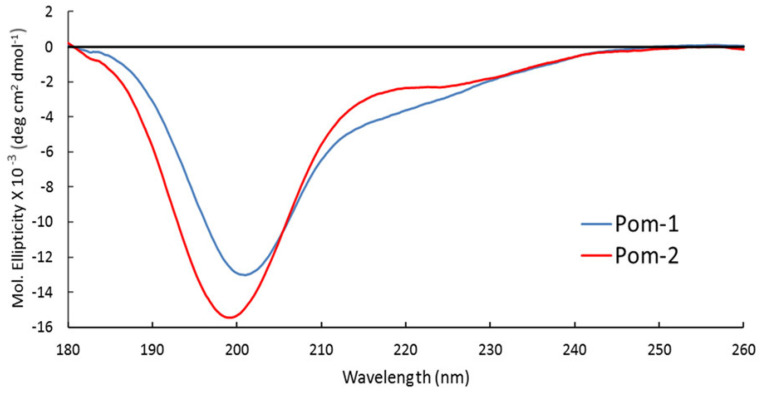
Circular dichroism (CD) spectra of peptides Pom-1 (blue) and Pom-2 (red) at 30 µM concentration in H_2_O.

**Figure 4 biomolecules-10-01473-f004:**
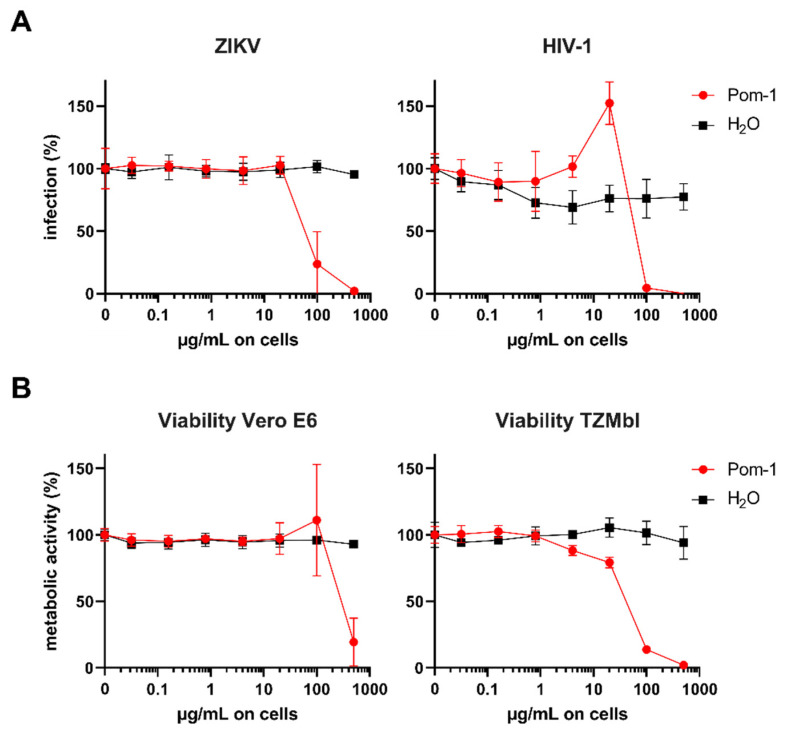
Effect of Pom-1 on Zika virus (ZIKV) and Human immunodeficiency virus 1 (HIV-1) infection. (**A**) Vero E6 or TZMbl cells were treated with Pom-1 or water and infected with ZIKV MR766 or HIV-1 NL4-3, respectively. Two days later, infection rates were quantified using cell-based ZIKV immunodetection or β-galactosidase reporter assay. Values were corrected for the background signal derived from uninfected cells. (**B**) Vero E6 or TZMbl cells were treated with Pom-1 or water according to the infection protocol. Two days later, metabolic viability was assessed by MTT-based assay. Values of untreated controls were set to 100% viability. Data in (**A**,**B**) are normalized to values in the absence of the respective compound and represented as average values obtained from two independent experiment in triplicates ± standard deviations.

**Figure 5 biomolecules-10-01473-f005:**
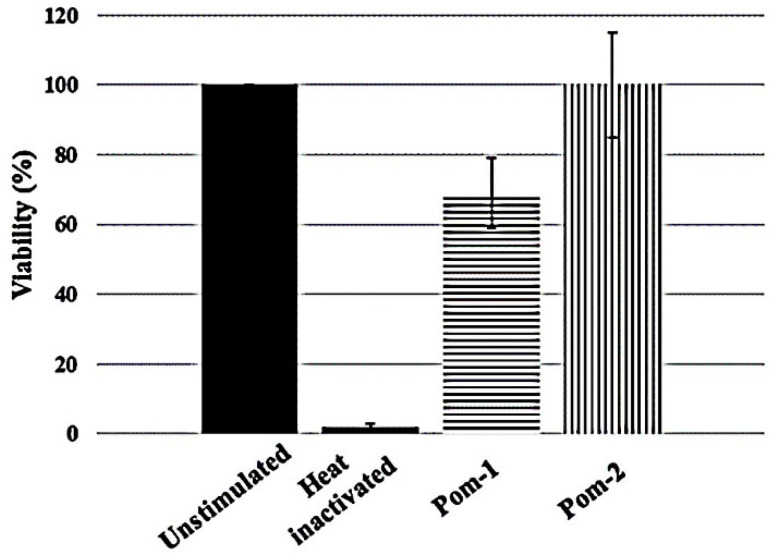
Cytotoxic effects of *P. poeyana* peptides at 100 µg/mL on primary human macrophages. Experiments were performed in triplicates, and the results are shown as mean values. Standard deviation is shown as vertical bars.

**Table 1 biomolecules-10-01473-t001:** Structure evaluation of the six predicted models for Pom-1 and Pom-2.

Model	MolProbity Score	Clash Score	Poor Rotamers	Favored Rotamers	Ramachandran Outliers	Ramachandran Favored	Cβ Deviations	Bad Bonds	Bad Angles
Pom-1_1	2.72	5.9	2/22	17/22	2/32	27/32	3/27	0/246	2/326
Pom-1_2	3.1	7.8	7/22	14/22	2/32	29/32	1/27	0/246	0/326
Pom-1_3	2.98	3.9	3/22	15/22	3/32	18/32	0/27	0/246	1/326
Pom-1_4	2.86	2.0	7/22	11/22	2/32	25/32	1/27	0/246	3/326
Pom-1_5	1.99	0.0	4/22	16/22	0/32	29/32	2/27	0/246	0/326
Pom-2	1.41	0.0	2/24	21/24	1/30	29/30	1/31	0/245	0/328

**Table 2 biomolecules-10-01473-t002:** Secondary structure content of peptides Pom-1 and Pom-2 in H_2_O.

Peptide	Helix (%)	Sheets (%)	Turns (%)	Random (%)
Pom-1	22.2	8.9	20.8	48.1
Pom-2	21.0	2.8	21.7	54.5

**Table 3 biomolecules-10-01473-t003:** Agar diffusion test of Gram-positive and Gram-negative bacterial species against *Pomacea poeyana* peptides.

Bacterial Species	Pom-1						Pom-2					
Concentration (µg/mL)	5	10	20	30	40	50	5	10	20	30	40	50
*Pseudomonas aeruginosa* ATCC27853	0.4	0.55	0.65	0.7	0.8	0.85	-	-	-	0.2	0.2	0.45
*Lysteria monocytogenes* ATCC BAA-679/EGD-e	0.25	0.35	0.35	0.35	0.4	0.45	-	-	-	0.15	0.3	0.4
*Klebsiella pneumoniae* ATCC 70063	-	-	0.3	0.3	0.4	0.43	-	-	-	-	-	-

**Table 4 biomolecules-10-01473-t004:** Values of the 50% cytotoxic concentration (CC_50_), half maximal inhibitory concentration (IC_50_) and selectivity index (SI).

Virus	ZIKV (Vero E6)	HIV-1 (TZMb1)
**IC_50_**	88.05	80.58
**CC_50_**	438.00	40.16
**SI**	4.97	0.50

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
