# Peer review of "New Antibacterial Peptides from the Freshwater Mollusk Pomacea poeyana (Pilsbry, 1927)"

_biomolecules, 2020, doi:10.3390/biom10111473_

Round 1

Reviewer 1 Report

Molecules
Manuscript Number: biomolecules-937866-
Title “New antibacterial peptides from the freshwater  mollusk Pomacea poeyana (Pilsbry, 1927)”, authors Melaine González-García, Armando Rodríguez, Annia Alba et al.
In this article, the authors present the results of a very interesting topic, Antimicrobial Peptides (AMP), with antimicrobial activity against a wide range of pathogens.
The presence of antimicrobial peptides in the freshwater mollusk Pomacea poeyana. 535 was demonstrated for the first time.
The authors present the results of characterization of 26 peptide fractions of the Cuban freshwater snail P. poeyana (Pilsbry, 1927), identifying 37 different peptides.
Many modern methods of analysis have been used, as the amino acid sequences of the peptides have been determined by nanoLC-ESI-MS-MS, the 3D structure of Pom-1 has been predicted using modeling in the QUARK de novo protein structure prediction server.
The structures and properties of two potential antimicrobial peptides were determined by bioinformative forecasting.
Two peptides: Pom-1 - a fragment of Closticin 574 isolated from the bacterium Clostridium tyrobutyrium, and Pom-2 - a fragment of cecropin D-like peptide isolated for the first time from the hemolymph Galleria mellonella, are very interesting with antifungal activity against
Candida sp., antibacterial against Gram-negative species P. aeruginosa and K. pneumoniae and Gram-positive species L. monocytogenes.

The results are very interesting, but some additional information and discussion need to be add:
1. Similar peptides have been identified in other mollusc organisms as snails Helix lucorum and their similarity should be discussed.
  1. Peptides with different amino acid sequences have been isolated from other organisms as Rapana and helix and should be compared.
  2. Peptide structures from other mollusks have been predicted. Is there a similarity between Pom-1 and Pom-1?
  3. What is the similarity with other peptides that exhibit antibacterial properties?
·       P. Dolashka, A. Dolashki, et al.. Antimicrobial activity of peptides from the hemolymph of Helix lucorum snails. Int.J.Curr. Microbiol. App. Sci 4, 4, 1061-1071 (2015)

Author Response

We have added the points proposed by reviewer 1 and discussed them in our new manuscript (discussion section marked in yellow). We have also considered the proposed reference [30].

Point 1 and 3. Similar peptides have been identified in other mollusc organisms as snails Helix lucorum and their similarity should be discussed. Peptides with different amino acid sequences have been isolated from other organisms as Rapana and helix and should be compared.

Lines: 451-456: Pom-1 and Pom-2 do not belong to the group of prolin-rich AMPs from mollusks [30] nor they contain disulfide bridges like the majority of AMPs from mollusks many of them belonging to the family of defensins or defensin-like peptides and larger proteins [31]. However, a few AMPs from mollusks exist that do not have cystein residues, e.g. the novel 55 amino acid peptide cgMolluscidin (extracted from the pacific oyster Crassostrea giga) [32], the piscidin-like AMPs (derived from Oreochromis niloticus) [33] and others.

Point 4: Peptide structures from other mollusks have been predicted. Is there a similarity between Pom-1 and Pom-2 ?

Lines 469-472: The predicted structure of Pom-1 and Pom-2 is highly helical in nature, with many positively charged residues and a similar tertiary structure arrangement of two alpha-helix joined by a short loop, a common feature in some antimicrobial peptides from insects such as sarcotoxin IA and Pd [37], melittin [38], papiliocin [39] and cecropins [40].

Point 5: What is the similarity with other peptides that exhibit antibacterial properties ?

Lines 520-527: Pom-1 and Pom-2 share structural characteristics with some antibacterial peptides like sarcotoxins, papiliocin and melittin, mentioned above. They have demonstrated antibacterial activity against both Gram-positive and Gram-negative bacteria [48]. In this sense, Lee and collaborators demonstrated that papiliocin have bacterial cell selectivity with no cytotoxicity against NIH 3T3 cells and related this action with their high cationicity [48]. Other peptides with antibacterial activity are the cathelicidin family, which showed their strong effects on a large number of drug-resistant strains clinically isolated, even super-resistant bacteria. Its antibacterial activity is attributed to its N-terminal amphipathic α-helix [49].

Reviewer 2 Report

Review of the manuscript biomolecules-937866-

Title: New antibacterial peptides from the freshwater
3 mollusk Pomacea poeyana (Pilsbry, 1927)

AMPs have been demonstrated to as potential alternative to the antibiotic treatment as they show a broad-spectrum of antimicrobial activities including anti-bacteria, anti-fungi, anti-viruses, and anti-cancers, and they could possibly overcome bacterial drug-resistance. In report several peptides from Cuban freshwater snail Pomacea poeyana (Pilsbry, 1927) were isolated and using bioinformatics structural analysis two of them were assign as AMPs. These peptides were synthetized and their biological activity on various pathogen was confirmed.

I would appreciated confirmation of theoretically suggested structure by experimental approach using for example by circular dichroism or NMR spectroscopy. The authors in discussion speculates role of amphipathicity and hydrophobicity and also about possible interaction with discrete components of bacterial membranes. This discussion could be also supported by some experimental data, but on the other hand such work is behind scope of this report.  However the experimental data could help describe details of the mechanism of action of studied AMPs and probably also to suggest the changes in their primary structure to improve their activity.

Author Response

As suggested by reviewer 2, experimental CD data have been included in the results and discussion section (new Figure 3 and Table 2). These obtained experimental structural data principally support our structural models obtained from the prediction software. We have amended this comparison between the experimental obtained data and the predicted structures in the manuscript and have discussed it with respect to a possible mechanism of action as proposed by reviewer 2. The corresponding sections are marked in yellow in the revised manuscript.

Round 2

Reviewer 2 Report

In abstract the authors wrote

The CD spectra analysis of POM-1 and POM-2 predicted for both structures a high random coil content and a very low β-sheet content. Pom-1 showed the highest content of both α-helix and β-sheet which is in accordance to other described AMPs displaying a disordered structure in water but adopting a helical conformation in presence of bacterial membrane analogues….

however the presented CD spectra are measured only in water and no membrane mimicking environment was used, thus this preposition from my point of view should be reformulated. This would then be in agreement with those what is written in discussion, where based on comparison with structural similar peptide papilocin and already presented results in literature (cit 18) author suggested the possible formation of the a-helical conformation for POM1 and POM-2 peptides. No such experiment mentioned in abstract was attempted.

andom coil content and a very low β-sheet content. Pom-1 showed the highest content of both α-helix and β-sheet which is in accordance to other described AMPs displaying a disordered structure in water but adopting a helical conformation in presence of bacterial membrane analogues….

Author Response

Reviewer 2: In the abstract one sentence was mistakenly written. As recommended by the reviewer we have rewritten this sentence to clarify the point that the presented CD spectra were measured in water and that structural predictions have made in presence of micelles (text marked in yellow). The conclusions now are in agreement with the discussions section.